# Cooperative Efficiency Evaluation System for Intelligent Transportation Facilities Based on the Variable Weight Matter Element Extension

Kailei Li [1,*], Han Bai [1,2,*], Xiang Yan [1], Liang Zhao [1,2] and Xiuguang Wang [1,2]

[1] School of Transportation and Logistics Engineering, Shandong Jiaotong University, Jinan 250300, China
[2] Shandong Key Laboratory of Smart Transportation (Preparation), Jinan 250101, China
* Correspondence: likailei2021@163.com (K.L.); baihan@sdjtu.edu.cn (H.B.)

**Abstract:** In order to effectively evaluate the cooperative efficiency of intelligent transportation facilities, a structural model of four cooperative development elements, including functional cooperative, operational cooperative, information cooperative, and operation cooperative, is constructed with the guidance of system coordination and a cooperative efficiency evaluation system is established based on it. Then, a dynamic efficiency evaluation model based on variable weight and matter-element extension method was constructed to describe the cooperative efficiency of intelligent transportation facilities and analyze the cooperative efficiency of key road sections in the Jinan area as an example. The results show that of the ten sections, two are in poor performance status, three are in good performance status, and five are in excellent performance status. The four indexes of vertical cooperative construction, functional scheduling level, information element completeness, and multi-departmental information integration level have the most significant impact on facility cooperative efficiency and are the most sensitive; the three indexes of plan executability, functional ease of upgrading, and space–time alignment rate have the most negligible impact on facility cooperative efficiency and are the least sensitive.

**Keywords:** traffic engineering; traffic facilities; evaluation index system; cooperative development; variable weight theory; matter element analysis

## 1. Introduction

With the development of new technologies such as intelligent networked vehicles and traffic big data, the intellectual posture of facilities continues to strengthen, and the communication between facilities and other individuals deepens. In addition, the scale of fixed assets and physical stock of transportation facilities continue to rise, posing new challenges to the operation and maintenance management [1] and traffic efficiency [2] of transportation systems. Various intelligent transportation facilities in the city are not separate individuals, strengthening the organic combination with other facilities, sharing functions and information between facilities [3,4], forming a "1 + 1 > 2" effect, is the future development trend of facility construction. Since the cooperative development of intelligent transportation facilities in China is still in the exploratory stage, coupled with the genetic problems of not considering the needs of urban transportation development, a large number of duplicate constructions [5–7]", information silos [8]", insufficient development of the facilities themselves [9], and "fragmentation" of policies and regulations. In the intelligent transportation system, the problem of low level of cooperative development in terms of functions, information, business and operation, and maintenance occurs in the participating subjects such as transportation and joint departments, operation and maintenance personnel, and service subjects, which makes it challenging to meet the increasingly diversified transportation needs. The uneven spatial distribution of facilities [10] also suffers from an imbalance between supply and demand. Therefore, conducting a

cooperative effectiveness evaluation of the facility is necessary. The cooperative efficiency evaluation is essential for its demonstration and development stages. It can reflect the actual situation to the higher authorities, anticipate advances such as resource utilization level [11] and energy loss [12], provide data support for subsequent planning, and also coordinate the multiple facilities at lower levels, identify deficiencies in time, and provide the theoretical basis for the development and exploitation of cooperative effectiveness of each facility.

As of the significant impact of the cooperative efficiency of facilities on transportation performance, more studies have started to explore the deeper meaning of facilities, mainly focusing on aspects such as facility construction project issues, facility operation and maintenance management, facility adaptability, and facility wisdom. However, most of them only target a particular subsystem and a specific type of intelligent facility in the ITS (Intelligent transportation system). Xiang et al. [13] established an industry-specific performance evaluation system for the implementation phase of transportation infrastructure PPP projects, while Hu et al. [14] combined entropy weight and COVA operator to eliminate the ambiguity of indexes and calculated the evaluation object and target closeness by improving the TOPSIS method to solve the problems in the partner selection process of transportation infrastructure PPP projects. Moreover, in engineering practice, promoting the transformation and upgrading of the construction and development of facilities [15] and improving the facility operation and maintenance management system [16,17] can further improve the reliability of facilities [18]. The analysis of the adaptability and accessibility of facilities after construction can also enhance traffic safety [19,20] and increase the satisfaction of residents. Ren et al. [21], through the influence of the accessibility of transportation facilities on the travel mode and distance of residents, argued that the construction of facilities needs to consider the needs of different residents fully. Furthermore, as the information society continues to develop, advanced concepts and technologies provide new ideas for the innovative development of facilities [22], and Wang [23] designed an intelligent monitoring and early warning system for traffic facility safety that can realize monitoring data collection, data transmission and preservation, and intelligent warning as a whole. Kerimov et al. [24] proposed a model for the operation of automatic traffic enforcement facilities and identified the main factors affecting the operation of this model. Moreover, Liu et al. [25] constructed a multi-dimensional evaluation system for the wisdom of public transportation infrastructure in Tianjin from the wisdom of facilities, management, and services. Furthermore, in the post-epidemic era, it is imperative to properly handle and respond to the impact of COVID-19 on the transportation sector [26]. While few studies on evaluating the cooperative efficiency of intelligent transportation facilities have been conducted, Kossov et al. [27] studied the assessment of the effectiveness of transportation facilities. However, the study mainly focused on unilateral transportation effectiveness without considering the indexes of the cooperative development of urban intelligent transportation facilities and did not involve the quantitative evaluation and grade classification of the effectiveness indexes. Given this, this study intends to construct an index system for evaluating the cooperative efficiency of intelligent transportation facilities and build an evaluation model, which is the first breakthrough of this study.

When studying the problem of cooperative efficiency of facilities, the selection of evaluation methods is the key, and the matter-element extension method is a standard evaluation method that has been widely used in the evaluation of soil and water resources quality [28], psychological stress evaluation [29], and traffic evaluation [30,31]. The matter-element extension method solves real-world contradictions and incompatibilities by analyzing the ambivalence among subsystems of decision objects through correlation functions [32]. Evaluating the cooperative efficiency of intelligent transportation facilities is a multi-index decision process. There is a problem of incompatibility of the evaluation results of a single index, which can be used to evaluate the level of cooperative efficiency of intelligent transportation facilities by the matter-element extension method. The constant weight method makes the weights of evaluation indexes fixed at any point in time and in any

scenario, and evaluating the cooperative efficiency of facilities should highlight the degree of influence of a key index. And when this index is in an extreme situation, the constant weight method is not responsive enough to the dynamic and timely condition of the whole transportation facility system, and the dynamic weight can help to solve this problem [33]. The weight of the evaluation index changes with the change of the gap between the performance play state value and the performance excellence interval, and the weight can better reflect the dynamic impact of each index on the intelligent transportation system to help traffic management to identify the key influencing factors and assist in decision making. Scholars have applied the concept of dynamic weights to the risk of road construction projects [34]. However, the research is fixed for the same index with different levels of punishment out of the excellent interval and the same level of punishment for the same degree of different importance index out of the excellent interval, which has a particular gap with the management practice. Due to this, this study combines dynamic power with the matter element to possible method to judge the development and change trend of evaluation objects by correlation size, characterize the dynamic change process of complex systems, realize dynamic efficiency evaluation, and improve the objectivity and scientificity of evaluation level determination, which is the second breakthrough of this study.

In this study, guided by the system coordination theory and combined with the characteristics of the development of intelligent transportation facilities, a structural model of the cooperative development elements of facilities with four degrees of functional cooperation, information cooperation, business cooperation, and operation cooperation is proposed, and an evaluation index system of cooperative effectiveness is constructed based on the structural model. Subsequently, the evaluation model was established by combining the variable weight theory and the matter-element extension method, and the case study of Jinan key road Jingshi Road was used for verification. To make up for the deficiencies of the existing constant weight research methods, improve the sustainable development of facility efficiency and enrich the research on the evaluation of the cooperative efficiency of intelligent transportation facilities in the theoretical aspect, and guide the construction of urban transportation facilities operation and maintenance in the practical element.

## 2. Materials and Methods

### 2.1. Study Area

This study was conducted on Jingshi Road, the main road in Jinan, Shandong Province, China. With a length of about 90 km and a width of about 60 m, Jingshi Road is a relatively well-constructed facility that runs across the east–west central axis of the Jinan area against the south line and is one of the longest urban roads in China, undertaking long-distance vehicle traffic flow transportation across sections (Figure 1). There are several squares, schools, hospitals, government, and other public places in the road section, and there are long-term traffic jams and high accident rates, so the evaluation of the cooperative effect of facilities for this regional road section is well-represented.

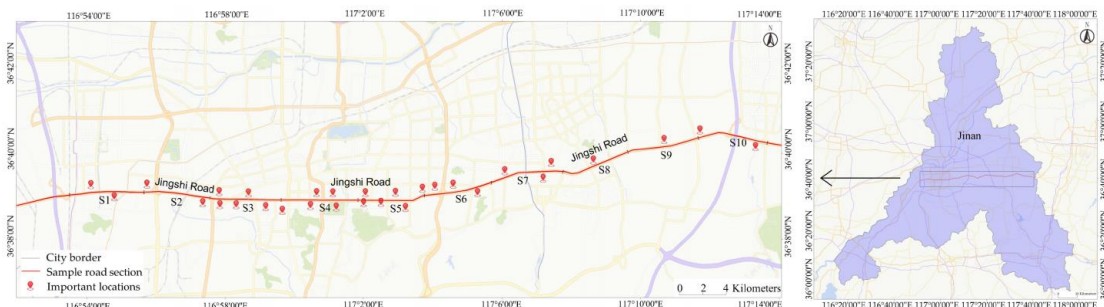

**Figure 1.** Map of the study area showing the Map of the study area shows the area of Jinan, Shandong Province, China, and highlights the surveyed roads and essential locations on both sides of the roads in Jinan. The surveyed road sections start at Qizhou Road and end at the Gangxi Interchange, dividing the road into ten sections S1, S2, S3, S4, S5, S6, S7, S8, S9, and S10. Source: Author's elaboration.

## 2.2. Collection and Preparation of Data

The study area has a wide variety of facilities with complex functions, and in the principle of reasonableness and fairness, we collected data on representative facilities such as multifunctional electric police, traffic bayonet, major speed measurement equipment, capture equipment, signal lights, and dynamic guidance facilities, and selected some critical sections of the Jingshi Road, with each section being about 3 km for a total of 30 km and ten sections for analysis and research. The majority of the data for this study came from the Jinan Statistical Yearbook, Jinan Lixia District Statistical Yearbook, China Urban Construction Statistical Yearbook, Jinan Urban and Rural Transport Bureau statistics, Jinan public open data, and statistics and analysis of representative facilities data in ten sections of Jingshi Road. Data from the Jingshi Road Sample were thoroughly processed using Origin, EXCEL, SPSS, and STATA (latest versions) to comprehensively assess the degree of the Jingshi Road facility cooperative efficiency. Subsequently, key informant interviews and questionnaires were conducted in these sections. According to the constructed evaluation system, relevant experts and knowledgeable people are invited to interview and score the evaluation subjects. A total of 100 questionnaires were returned, and 97 (97%) of them were completed and considered valid, with an average score of the sample sections 76.5. Key informant interviews showed that the facility performance was well-maintained in the early stages of the construction of the facility, with strict control of road infrastructure standards, relatively high service levels, and was able to meet the expected requirements. With the renewal and replenishment of facilities, the problems of haphazard alteration, duplicate construction, and independent construction of multi-sectoral facilities have emerged, thus providing better evidence of the promising coordinated efficiency evaluation of facilities.

## 2.3. Establishment of a Cooperative Evaluation System for the Effectiveness of Intelligent Transportation Facilities

The American scholar Professor Harken [35] extends the cooperation theory from business management to developing anything. He believes all things and activities exist in order and disorder. In a particular case, the two states of order and chaos could switch to each other under external or internal action, and demand is what we consider cooperation. In addition, Ludwig Von Bertalanffy pointed out the famous "non-additive law", that is, in mathematics $1 + 1 = 2$, in systems theory $1 + 1 \neq 2$, the specific model is: $E = \sum\limits_{i=1}^{n} e_i + P$, where $E$ represents the overall function of the system, $e_i$ represents the function of the components of the system $i$, $P$ represents the structural function formed by the connection of the elements. We can see through the model: $P > 0$ that the "whole is greater than the sum of the parts", that is, the overall cooperative effect; $P < 0$ that the "whole is less than the sum of the parts", that is, the overall reduction effect [36]. Cooperative development of facilities is a kind of "integrity", "comprehensive", and "endogeneity" synchronous development of the convergence, which is the end of the urban transportation system of intelligent facilities to adapt to each other and collaborate to promote, coupled with the process of synchronous development of the virtuous cycle of posture. The purpose of the research on the cooperative development of facilities is to reduce the adverse effects within the urban transportation system, improve the overall effectiveness of the facilities, and indirectly improve the overall output effectiveness of the urban transportation system. Therefore, by constructing a structural model of the elements of cooperative development of facilities in four modules, such as functional cooperation, business cooperation, information cooperation, and operation cooperation (Figure 2), we further construct an index system of evaluation indexes (Table 1).

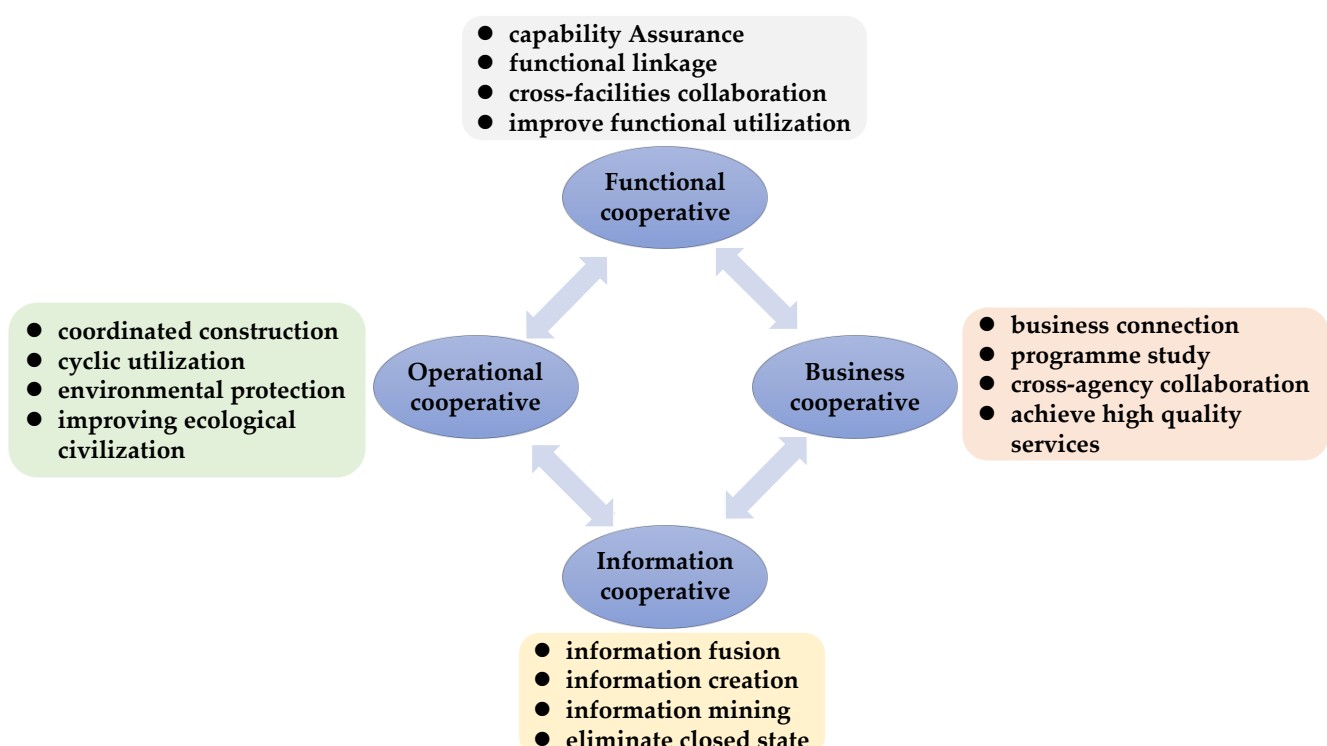

**Figure 2.** Structural model of facility cooperative development elements.

**Table 1.** Hierarchy of the evaluation system.

| Target Layer | Criterion Layer | Index Layer | |
|---|---|---|---|
| | | **Index** | **Type** |
| Functional cooperative, $B_1$ | Functional integrity, $C_1$ | Functional coverage, $D_1$ | + |
| | | Functional peak level, $D_2$ | + |
| | | Functional reproducibility, $D_3$ | - |
| | Function expansion, $C_2$ | Function guarantee level, $D_4$ | + |
| | | Functional upgrading level, $D_5$ | + |
| | | Interface reservation level, $D_6$ | + |
| | | Functional variability, $D_7$ | + |
| | Functional sharing breadth, $C_3$ | Function scheduling level, $D_8$ | + |
| | | Functional combination effect, $D_9$ | + |
| Business cooperative, $B_2$ | Project level, $C_4$ | Plan autonomy, $D_{10}$ | + |
| | | Plan executability, $D_{11}$ | + |
| | Option evaluation level, $C_5$ | Program plan efficiency, $D_{12}$ | + |
| | | Option evaluation accuracy, $D_{13}$ | + |
| | | Option evaluation efficiency, $D_{14}$ | + |
| Information cooperative, $B_3$ | Information fusion level, $C_6$ | Space-time registration rate, $D_{15}$ | + |
| | | Information interactivity, $D_{16}$ | + |
| | | Multi-sectoral information fusion level, $D_{17}$ | + |
| | Information evaluation level, | Traffic environment evaluation, $D_{18}$ | + |
| | | multi-department mobility evaluation, $D_{19}$ | + |
| | | Control ability evaluation, $D_{20}$ | + |
| | | Information evaluation efficiency, $D_{21}$ | + |
| | Information expression level, $C_8$ | Information updating rate, $D_{22}$ | + |
| | | Completeness of information elements, $D_{23}$ | + |
| | | Information display accuracy, $D_{24}$ | + |
| | Information forecast level, $C_9$ | Facilities status, $D_{25}$ | + |
| | | Road condition alarm, $D_{26}$ | + |
| | | situation prediction efficiency, $D_{27}$ | + |

**Table 1.** *Cont.*

| Target Layer | Criterion Layer | Index Layer | |
|---|---|---|---|
| | | Index | Type |
| | Repeated construction level, $C_{10}$ | Vertical cooperative construction, $D_{28}$ | - |
| | | Facility consolidation level, $D_{29}$ | + |
| | Reliability level, $C_{11}$ | Annual failure rate, $D_{30}$ | - |
| | | Mean time between failure, $D_{31}$ | - |
| Operational cooperative, $B_4$ | | Recoverable level, $D_{32}$ | + |
| | | Facilities replacement cycle, $D_{33}$ | + |
| | Resources utilization level, $C_{12}$ | Cycle level of facilities, $D_{34}$ | + |
| | | Spatial intensification level, $D_{35}$ | + |
| | | Renewable resources utilization level, $D_{36}$ | + |

Note: In the table,"+" means that the index is conducive to cooperative efficiency, and "-" means that the index is not conducive to cooperative efficiency.

Among them, the essence of functional cooperation is to reach a consensus on the optimal configuration of different functions among various facilities, alternative, complementary, and enhancement measures to improve the efficiency of the use of functional resources, optimize the tasks of multiple facilities from a global perspective, form cooperation of scattered functions, and promote the utilization of resources and the improvement of overall efficiency. The core of operational cooperation is to provide seamless services, "loop together", and reduce friction, especially in urban road conditions that are diverse and personalized, to improve the operational level and response speed, to achieve the goal of smooth traffic flow, to provide high-quality services, and to achieve the improvement of overall efficiency within the system. The information cooperation not only enables the free flow of information among facilities but also promotes the innovation and improvement of the intelligent transportation system by mining and analyzing traffic data, acquiring confidential information, eliminating closed status, adjusting the allocation of resources promptly, and the interconnection of information among subsystems and facilities makes urban transportation smooth, safe, and efficient. In addition, operation cooperation as the basis of the cooperative development of facilities, the overall coordination of facility construction, the level of recycling and reliability of operation, and ecological environmental protection [37–39] are fundamental to guarantee the essential efficiency of facilities.

*2.4. Determination of Index Weights*

2.4.1. Index Normalization

When the index value exceeds the range of the nodal domain, using the correlation function calculation, there is a denominator of zero, which cannot be evaluated typically. In addition, to overcome the maximum subordination problem that cannot reflect the ambiguity of the object to be evaluated in some cases of its boundaries and cause deviation of the evaluation results [40], the efficacy coefficient method uses the standardization of evaluation indexes. The greater the positive index value, the better the efficiency, which uses Equation (1) for standardization. The smaller the value of the negative index, the better the efficiency, which uses Equation (2) for standardization. The efficiency interval of each index to set as excellent, good, medium, and poor.

$$x_i = \begin{cases} \left(\frac{M_i}{X_i}\right) \times 0.85 & X_i \geq M_i \\ 1 - \left(\frac{M_i}{X_i}\right) \times 0.15 & X_i < M_i \end{cases} \tag{1}$$

$$x_i = \begin{cases} 1 - \left(\frac{M_i}{X_i}\right) \times 0.15 & X_i \geq M_i \\ \left(\frac{M_i}{X_i}\right) \times 0.85 & X_i < M_i \end{cases} \tag{2}$$

In Equations (1) and (2), $x_i$ is the standardized value, $X_i$ is the actual value of the evaluation index, and $M_i$ is the excellent and non-optimal critical line level of the evaluation index.

2.4.2. Determine the Constant Weight

The paper uses the order relation analysis method to calculate the constant weights of indexes. The method improves the AHP, which optimizes the problems of extensive computation and inconsistent judgment matrix in the AHP. The specific steps are:

(1) Determine the order relation. We are ranking the evaluation indexes in order of relative importance;

(2) Determining the importance of the index. We invited several experts in the field of transportation to determine the importance level ratio of adjacent indexes; and

(3) Calculation of index weights. Based on the principle that the weights of all indexes sum to 1, the constant weights are calculated based on Equation (3).

$$
\begin{cases}
w_i = (1 + \sum_{i=2}^{n} \prod_{i=n}^{n} r_i)^{-1} & (i = 2, 3, \cdots, n-1, n) \\
r_i = w_i / w_{i-1}
\end{cases} \tag{3}
$$

In Equation (3), $w_i$ is the constant weight variable and $r_i$ is the ratio of importance degree of adjacent indexes.

2.4.3. Determination of Variable Weight

The design of variable weights adjusts the weights of indexes below the standard level to consider facility cooperative effectiveness evaluation dynamics. It reduces the value of the comprehensive evaluation indexes to punish the indexes below the standard level. The calculation principle is that the variable weight vector $W_{i(X)} = (W_{1(X)}, W_{2(X)}, \cdots, W_{m(X)})$ is the normalized Hadamard product of the constant weight vector $w$, and the state variable weight vector $S$, and the state variable weight vector $S$ is the gradient vector of the equilibrium function, as shown in Equation (4).

$$
W_{i(X)} = \frac{W_i S_{i(X)}}{\sum_{i=1}^{m} W_i S_{i(X)}} \tag{4}
$$

Different punishment is applied to indexes with different weights of constant weights when they deviate from the same degree of the threshold, and different punishment is applied to the same index that deviates from different levels of the excellent interval of effectiveness by setting different punishment intervals. Therefore, based on the research of Zhao [41], the structure is improved and extended, as shown in Equation (5).

$$
S_{i(X)} = \begin{cases}
1 & \beta_1 < x_i \le 1 \\
e^{-\alpha_i(\beta_1 - x_i)} & \beta_2 < x_i \le \beta_1 \\
e^{-2\alpha_i(\beta_1 - x_i)} & \beta_3 < x_i \le \beta_2 \\
e^{-3\alpha_i(\beta_1 - x_i)} & x_i \le \beta_3
\end{cases}, \quad \alpha_i = \alpha \times \left(\frac{w_i}{w_0}\right) \tag{5}
$$

In Equation (5), $\beta_1$ is the critical line of excellent and non-excellent performance after the standardization of evaluation indexes, $\beta_2$ is the threshold of good and medium performance after the standardization of evaluation indexes, $\beta_3$ is the threshold of medium and poor performance after standardization of evaluation indexes, and after normalization, $\beta_1$, $\beta_2$, $\beta_3$ are 0.913, 0.861, and 0.827, respectively. $x_i$ is the measured value of the index $i$. When $\beta_1 < x_i \le 1$, the index is in the excellent performance range, and the $i$ index is without punishment. When $\beta_2 < x_i \le \beta_1$, the index value is in a suitable performance interval, the corresponding weight of b becomes more significant, and the $i$ index with punished. When $\beta_3 < x_i \le \beta_2$, the index value is in the middle range of efficiency, the corresponding weight of b increases rapidly, and the $i$ index with more severe punishment. When $x_i \le \beta_3$, the index value is in the performance difference interval, the corresponding weight of $i$ increases rapidly, and the index of $i$ with severe punishment. $\alpha$ is the base penalty factor, determined by the expert empirical method $\alpha = 0.81547$ [42]. The heavier the penalty below the threshold level, the more significant the negative impact on the evaluation results.

*2.5. Matter-Element Extension Evaluation Model*

2.5.1. Determine the Evaluation Matter-Element

Given the cooperative efficiency of facilities $N$, the quantity value $c$ about the feature $v$, the ordered triad $R = (N, c, v)$ is used as the essential element to describe the cooperative efficiency of facilities, referred to as the matter-element; the effectiveness $N$, the feature $c$, and the quantity value $v$ are called the matter-element triad. Equation (6) shows that the matter element $N$ of the efficiency comprises $n$ characteristic indexes $c_1, c_2, \cdots, c_n$ that reflect the efficiency with matching values $v_1, v_2, \cdots, v_n$.

$$R = (N, C, V) = \begin{bmatrix} N & c_1 & v_1 \\ & c_2 & v_2 \\ & \vdots & \vdots \\ & c_n & v_n \end{bmatrix} \tag{6}$$

2.5.2. Determining Classical Domains, Node Domains, and Evaluation Objects

The classical domain $R_{zi}$ is the range of values of the index to be evaluated, the joint domain $R_P$ represents the total range of quantitative values of the characteristics of cooperative efficiency (efficiency indexes) across all evaluation grades, and the matter element to evaluate $R_T$ is the actual data of the cooperative effectiveness evaluation index.

$$R_{zi} = (N_{zi}, c_i, V_{zi}) = \begin{bmatrix} N_z & c_1 & (a_{z1}, b_{z1}) \\ & c_2 & (a_{z2}, b_{z2}) \\ & \vdots & \vdots \\ & c_n & (a_{zn}, b_{zn}) \end{bmatrix}, \tag{7}$$

$$R_p = (N_p, c_i, V_{pi}) = \begin{bmatrix} N_p & c_1 & (a_{p1}, b_{p1}) \\ & c_2 & (a_{p2}, b_{p2}) \\ & \vdots & \vdots \\ & c_n & (a_{pn}, b_{pn}) \end{bmatrix}, R_T = \begin{bmatrix} T & c_1 & v_1 \\ & c_2 & v_2 \\ & \vdots & \vdots \\ & c_n & v_n \end{bmatrix} \tag{8}$$

where $R_z$ is the classical domain element of the cooperative efficiency of facilities, $R_p$ is the nodal domain element, and $R_T$ is the element to evaluate. $V_{pi}$ is the range of $c_i$ and $v_{zi} = (a_{zi}, b_{zi})$ is the range of $v_i$. In addition, $N_z$ is the $z$th evaluation grade ($z = 1, 2, \cdots, m$), $c_i$ is the $i$th evaluation index, $v_{zi} = (a_{zi}, b_{zi})$ is the range of the $i$th evaluation index at the $z$th evaluation grade, $a_{z(p)i}$ and $b_{z(p)i}$ are the upper and lower limits of the range of values, respectively, and $v$ is the value of $T$ about $c_i$.

2.5.3. Determine the Close Degree

A theoretical analysis of the close degree criterion instead of the criterion based on maximum membership grade, the proposed asymmetric closeness equation ($p = 1$) [43] collates to obtain Equation (9).

$$N = 1 - \frac{1}{n(n+1)} \sum_{i=1}^{n} D w_i \tag{9}$$

where $N$ is the progress of a grade posting, $D$ is the distance from a point $x$ to the interval $X_0 = [a, b]$, and $w_i$ is the weight coefficient.

The comprehensive paste progress for each grade corresponding to the evaluation matter element is Equation (10).

$$N_z(t_0) = 1 - \frac{1}{n(n+1)} \sum_{i=1}^{n} D_z(v_i')w_i(X) \tag{10}$$

where $n$ is the total number of evaluation indexes, $D_z(v_i')$ is the distance between the evaluation element $v_i'$ and the classical domain $(a_{zi}', b_{zi}')$ under normalization, $D_z(v_i') = \left| v_i' - \frac{a_{zi}' + b_{zi}'}{2} \right| - \frac{(b_{zi}' - a_{zi}')}{2}$, and $w_i(X)$ is the weight coefficient of the evaluation index $v_i$.

### 2.5.4. Rating

By $N_z' = \max\{N_z(p_0)\}$, it can determine that the evaluation matter element belongs to the $z$ grade. In order to better judge the degree of bias of the evaluation element $R_T$ toward the adjacent rank, the eigenvalue of $R_T$ calculation shows in Equation (11).

$$z' = \frac{\sum\limits_{z=1}^{m} z \overline{N}_z(t_0)}{\sum\limits_{z=1}^{m} \overline{N}_z(t_0)} \tag{11}$$

where $z'$ is the characteristic value of grade variable of the evaluation matter element $R_T$, and $\overline{N}_z(t_0)$ is the normalized degree of each rank fit [44], $\overline{N}_z(t_0) = \frac{N_z(t_0) - \min\limits_z N_z(t_0)}{\max\limits_z N_z(t_0) - \min\limits_z N_z(t_0)}$.

## 3. Results

### 3.1. Weights of Assessment Index

The paper invites experts to assess the indexes relative importance and calculates each indexes constant weight according to Equation (1). Eleven experts were invited, three of whom were managers of the King's Road, two road safety experts, three experts who had been involved in the road project, and three traffic facility experts. The 36 indexes of the road section were evaluated according to the grade criteria. The variable weights were calculated using Equations (4) and (5) for the standardized evaluation means (Table 2).

**Table 2.** Results of the constant index weights and variable weights.

| Index | Constant Weight | Dynamic Weight | | | | | | | | | |
|-------|-----------------|--------|--------|--------|--------|--------|--------|--------|--------|--------|--------|
| | | S1 | S2 | S3 | S4 | S5 | S6 | S7 | S8 | S9 | S10 |
| $D_1$ | 0.0351 | 0.0324 | 0.0312 | 0.0341 | 0.0317 | 0.0325 | 0.0324 | 0.0330 | 0.0329 | 0.0317 | 0.0331 |
| $D_2$ | 0.0191 | 0.0177 | 0.0176 | 0.0187 | 0.0184 | 0.0194 | 0.0199 | 0.0198 | 0.0201 | 0.0203 | 0.0204 |
| $D_3$ | 0.0227 | 0.0245 | 0.0207 | 0.0234 | 0.0237 | 0.0234 | 0.0245 | 0.0269 | 0.0223 | 0.0235 | 0.0273 |
| $D_4$ | 0.0166 | 0.0162 | 0.0161 | 0.0178 | 0.0176 | 0.0176 | 0.0177 | 0.0164 | 0.0165 | 0.0161 | 0.0168 |
| $D_5$ | 0.0139 | 0.0138 | 0.0121 | 0.0120 | 0.0139 | 0.0139 | 0.0135 | 0.0135 | 0.0137 | 0.0135 | 0.0138 |
| $D_6$ | 0.0154 | 0.0124 | 0.0152 | 0.0152 | 0.0157 | 0.0162 | 0.0162 | 0.0151 | 0.0234 | 0.0151 | 0.0154 |
| $D_7$ | 0.0195 | 0.0232 | 0.0182 | 0.0191 | 0.0216 | 0.0221 | 0.0231 | 0.0197 | 0.0214 | 0.0203 | 0.0198 |
| $D_8$ | 0.0448 | 0.0433 | 0.0717 | 0.0432 | 0.0469 | 0.0467 | 0.0466 | 0.0472 | 0.0461 | 0.0857 | 0.0471 |
| $D_9$ | 0.0390 | 0.0393 | 0.0401 | 0.0403 | 0.0378 | 0.0399 | 0.0397 | 0.0397 | 0.0381 | 0.0378 | 0.0450 |
| $D_{10}$ | 0.0245 | 0.0326 | 0.0241 | 0.0257 | 0.0266 | 0.0234 | 0.0236 | 0.0261 | 0.0254 | 0.0214 | 0.0236 |
| $D_{11}$ | 0.0128 | 0.0148 | 0.0113 | 0.0134 | 0.0132 | 0.0128 | 0.0126 | 0.0135 | 0.0132 | 0.0132 | 0.0135 |
| $D_{12}$ | 0.0331 | 0.0318 | 0.0313 | 0.0389 | 0.0324 | 0.0342 | 0.0322 | 0.0297 | 0.0364 | 0.0281 | 0.0322 |
| $D_{13}$ | 0.0157 | 0.0178 | 0.0142 | 0.0142 | 0.0181 | 0.0163 | 0.0163 | 0.0178 | 0.0156 | 0.0123 | 0.0147 |
| $D_{14}$ | 0.0263 | 0.0224 | 0.0224 | 0.0254 | 0.0257 | 0.0261 | 0.0241 | 0.0223 | 0.0226 | 0.0180 | 0.0243 |
| $D_{15}$ | 0.0147 | 0.0144 | 0.0142 | 0.0135 | 0.0153 | 0.0138 | 0.0138 | 0.0176 | 0.0158 | 0.0131 | 0.0146 |
| $D_{16}$ | 0.0383 | 0.0348 | 0.0314 | 0.0362 | 0.0374 | 0.0375 | 0.0374 | 0.0413 | 0.0363 | 0.0314 | 0.0394 |
| $D_{17}$ | 0.0403 | 0.0411 | 0.0582 | 0.0397 | 0.0410 | 0.0425 | 0.0407 | 0.0438 | 0.0397 | 0.0476 | 0.0453 |
| $D_{18}$ | 0.0219 | 0.0215 | 0.0194 | 0.0217 | 0.0277 | 0.0219 | 0.0223 | 0.0186 | 0.0221 | 0.0153 | 0.0196 |
| $D_{19}$ | 0.0268 | 0.0325 | 0.0245 | 0.0241 | 0.0258 | 0.0226 | 0.0243 | 0.0241 | 0.0241 | 0.0220 | 0.0269 |
| $D_{20}$ | 0.0197 | 0.0208 | 0.0181 | 0.0184 | 0.0216 | 0.0201 | 0.0207 | 0.0192 | 0.0211 | 0.0171 | 0.0204 |
| $D_{21}$ | 0.0261 | 0.0243 | 0.0243 | 0.0263 | 0.0263 | 0.0252 | 0.0249 | 0.0243 | 0.0257 | 0.0226 | 0.0185 |
| $D_{22}$ | 0.0163 | 0.0181 | 0.0152 | 0.0171 | 0.0168 | 0.0169 | 0.0157 | 0.0157 | 0.0167 | 0.0141 | 0.0157 |
| $D_{23}$ | 0.0411 | 0.0417 | 0.0422 | 0.0437 | 0.0427 | 0.0387 | 0.0387 | 0.0425 | 0.0431 | 0.0617 | 0.0442 |
| $D_{24}$ | 0.0259 | 0.0224 | 0.0212 | 0.0283 | 0.0249 | 0.0263 | 0.0261 | 0.0214 | 0.0219 | 0.0212 | 0.0219 |

**Table 2.** *Cont.*

| Index | Constant Weight | Dynamic Weight | | | | | | | | | |
|-------|-----------------|------|------|------|------|------|------|------|------|------|------|
| | | S1 | S2 | S3 | S4 | S5 | S6 | S7 | S8 | S9 | S10 |
| $D_{25}$ | 0.0207 | 0.0275 | 0.0153 | 0.0204 | 0.0217 | 0.0180 | 0.0206 | 0.0231 | 0.0217 | 0.0176 | 0.0194 |
| $D_{26}$ | 0.0266 | 0.0251 | 0.0226 | 0.0258 | 0.0271 | 0.0270 | 0.0241 | 0.0264 | 0.0239 | 0.0194 | 0.0268 |
| $D_{27}$ | 0.0337 | 0.0327 | 0.0287 | 0.0324 | 0.0335 | 0.0357 | 0.0334 | 0.0319 | 0.0376 | 0.0283 | 0.0285 |
| $D_{28}$ | 0.0576 | 0.0582 | 0.0971 | 0.0580 | 0.0551 | 0.0582 | 0.0547 | 0.0553 | 0.0542 | 0.0802 | 0.0601 |
| $D_{29}$ | 0.0289 | 0.0297 | 0.0294 | 0.0293 | 0.0264 | 0.0274 | 0.0293 | 0.0297 | 0.0286 | 0.0279 | 0.0357 |
| $D_{30}$ | 0.0403 | 0.0421 | 0.0413 | 0.0406 | 0.0387 | 0.0421 | 0.0411 | 0.0421 | 0.0422 | 0.0386 | 0.0412 |
| $D_{31}$ | 0.0284 | 0.0277 | 0.0223 | 0.0282 | 0.0263 | 0.0278 | 0.0305 | 0.0298 | 0.0282 | 0.0256 | 0.0303 |
| $D_{32}$ | 0.0322 | 0.0310 | 0.0287 | 0.0367 | 0.0334 | 0.0334 | 0.0347 | 0.0318 | 0.0320 | 0.0278 | 0.0293 |
| $D_{33}$ | 0.0278 | 0.0242 | 0.0203 | 0.0268 | 0.0264 | 0.0238 | 0.0278 | 0.0261 | 0.0264 | 0.0243 | 0.0243 |
| $D_{34}$ | 0.0276 | 0.0216 | 0.0242 | 0.0262 | 0.0253 | 0.0281 | 0.0295 | 0.0274 | 0.0258 | 0.0244 | 0.0268 |
| $D_{35}$ | 0.0357 | 0.0337 | 0.0341 | 0.0336 | 0.0336 | 0.0368 | 0.0361 | 0.0360 | 0.0336 | 0.0347 | 0.0335 |
| $D_{36}$ | 0.0309 | 0.0327 | 0.0211 | 0.0316 | 0.0297 | 0.0317 | 0.0312 | 0.0312 | 0.0316 | 0.0281 | 0.0306 |

Herein, the facility efficacy evaluation degree of Jingshi Road is categorized as excellent (I), good (II), medium (III), and poor (IV). For the classic domain and section domain, the categories were established by consulting the Code for Design of Urban Road Traffic Facility (2019 edition), Ergonomics design guide for urban public transport facilities, Public transport-Service interface for real-time information relating to public transport operations-Part 5, and worldwide and national industry standards for each index. Based on the established effectiveness evaluation index system, the analysis took into account the actual situation of Jingshi Road and combined with the survey and analysis of the facilities and information processing as well as expert and staff consultation, the positive and negative index rating criteria were developed (Tables S1 and S2).

*3.2. Evaluation Results*

The rank distance, accessible degree, and eigenvalues of the ten sections (Table 3).

**Table 3.** Results of the effectiveness evaluation of the sample sections.

| Sections | Evaluation of the Accessible Degree of the Grade | | | | Maximum Proximity | Character Value | Grades |
|----------|------|------|------|------|-------------------|-----------------|--------|
| | $j = 1$ | $j = 2$ | $j = 3$ | $j = 4$ | | | |
| S1 | 0.1052 | 0.0473 | 0.0231 | 0.0193 | 0.1052 | 1.3025 | Excellent |
| S2 | 0.0763 | 0.0975 | 0.0165 | 0.1049 | 0.1049 | 2.5105 | Poor |
| S3 | 0.0861 | 0.0824 | 0.0237 | 0.0493 | 0.0861 | 1.9237 | Excellent |
| S4 | 0.1331 | 0.0756 | 0.0193 | 0.0128 | 0.1331 | 1.3998 | Excellent |
| S5 | 0.0877 | 0.1862 | 0.0760 | 0.0262 | 0.1862 | 1.9569 | Good |
| S6 | 0.1606 | 0.0741 | 0.0193 | 0.0127 | 0.1606 | 1.3455 | Excellent |
| S7 | 0.0873 | 0.1651 | 0.0346 | 0.0358 | 0.1651 | 1.7272 | Good |
| S8 | 0.1925 | 0.0729 | 0.0564 | 0.0639 | 0.1925 | 1.2436 | Excellent |
| S9 | 0.1085 | 0.0305 | 0.0425 | 0.1837 | 0.1837 | 2.9885 | Poor |
| S10 | 0.0387 | 0.1699 | 0.0275 | 0.0397 | 0.1699 | 2.0796 | Good |

S2 and S9 are in the efficiency difference interval, and in the analysis of each index within the sections, the evaluation means values of indexes $D_8$, $D_{17}$, and $D_{28}$ in S2 are 4.7, 3.9, and 7.4, respectively; the evaluation represents values of indexes $D_8$, $D_{17}$, $D_{23}$, and $D_{28}$ in S9 are 4.3, 4.1, 3.9, and 7.3, respectively, and the constant weight rankings of indexes $D_8$, $D_{17}$, $D_{23}$, and $D_{28}$ in the 36 indexes are 2, 4, 3, and 1. According to Equation (5), for those who are far away from the excellent interval, severe penalties are applied to make the index weight rise; for those who are too far away from the excellent interval, very severe penalties are to make the index weight rise rapidly. In management practice, facilities construction, operation, and maintenance require staff to combine the current roadway needs and the overall layout, coordinate planning and specific implementation, and strengthen the flexible scheduling of functions and information interaction and integration between facilities. For example, the installation of multiple monitoring facilities between an intersection by

different departments lacks integrated planning and scheduling, which seriously interferes with the overall efficiency of the facilities and therefore is also judged to be poorly effective in management practice, indicating that the evaluation level calculated by the study and the judgment of management practice is consistent.

S5 is in good effectiveness, analyzing each index of this section. However, the level of facility function interface reservation and upgradeability is lacking, the speed of information update is average, the execution of the traffic control plan is poor, and the level of old facility circulation has caused a particular impact on the cooperative development of the whole facility. However, the importance of these four indexes is relatively low, the effect on the overall efficiency is small, and the punishment on these indices is mild, so the section is in good effectiveness. Similarly, the peak business rate, guarantee function, and space–time alignment rate of the facilities in S7 have a particular gap from the excellent interval, and the higher level of duplication of functions, inadequate interface reservation, lower accuracy of traffic control scheme evaluation and longer update cycle of S10 have a particular impact on the effectiveness of the facilities. However, the importance of these indexes is relatively low, the impact on the overall efficiency is small, and the punishment on these indices is mild, so these two sections are in good efficiency intervals.

S1 effectiveness evaluation level in the effectiveness of excellent, analyzing the various indexes of this section, the positive index $D_{20}$ evaluation mean value is 4.7, the constant weight is in the 28th position, and the importance is relatively low. Nevertheless, the index value deviates from the effectiveness of excellent. It is slightly punishment, so its weight increases from 0.0197 to 0.0262, which barely impacts the efficacy of this section's facilities. Since there are no other risks in this section, the cooperative efficiency grade of the facilities in this section is judged to be excellent. The facility controls the road in a semi-autonomous state, with a human-machine approach, and road events often respond quickly, so it is judged to have an excellent performance rating in management practice. Similarly, the mean values of index evaluation of functional variability, traffic environment evaluation, facility status alarms, and control program evaluation accuracy of S3, S4, S6, and S8 deviate from the excellent range of effectiveness. However, the constant weights of these indexes are small, indicating that the indexes are less critical to the overall facility effectiveness and less punishing; coupled with the absence of other risks within the section, the efficiency of the facility is at an excellent grade.

## 4. Discussion

### 4.1. Sensitivity Analysis

The paper conducts a sensitivity analysis of the cooperative efficiency evaluation of the facilities to capture the pattern of change in effectiveness. When the value of one of the evaluation indexes changes and is in the effectiveness non-excellent, while the value of other evaluation indexes remains unchanged, the weight corresponding to that evaluation index will change, so the index sensitivity is judged by the change of the weight. Assuming that the status of the indexes of the sample section of the standard scenario is in an excellent performance, the positive indexes take the value of eight, and the negative indexes take the value of two. The index values are adjusted in the non-optimal interval of performance, and the positive indexes are decreasing step by step, and the negative indexes are increasing step by step, corresponding to scenarios 1–7. For example, $D_1$ is a positive index, assume that the initial value of the index is eight, and keep decreasing the value of the index until the index value is one, and analyze the influence trend of the index on the whole; $D_3$ is a negative index, assume that the initial value of the index is two, and keep increasing the value of the index until the index value is nine, and analyze the influence trend of the index on the whole, and so on for other indexes. The change rate of each index weight (Figure 3).

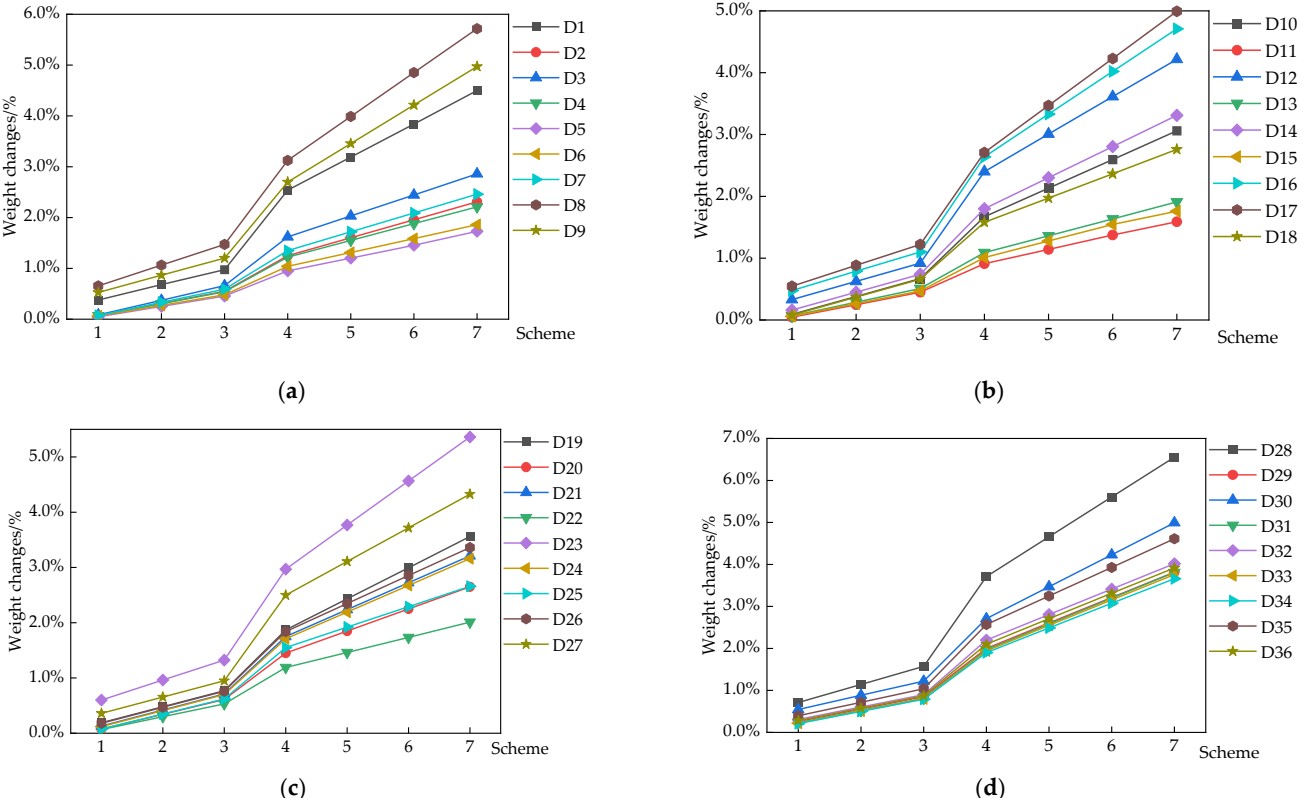

**Figure 3.** Change in the weighting of indexes deviating from the excellent interval. Changes in weights of indexes (**a**) $D_1$–$D_9$; (**b**) $D_{10}$–$D_{18}$; (**c**) $D_{19}$–$D_{27}$; (**d**) $D_{28}$–$D_{36}$.

As seen in the figure, the weight of each index gradually becomes more extensive as it deviates from the effectiveness excellence interval. The weight values of indexes $D_8$, $D_{17}$, $D_{23}$, and $D_{28}$ change the most, $D_8$, $D_{17}$, and $D_{23}$ index weights increased by 5.72%, 4.99%, and 5.36%, respectively, when the index value of $D_8$, $D_{17}$, and $D_{23}$ decreased from eight to one; the index weight of $D_{28}$ increased by 6.56%, when the index value increased from two to nine, indicating that these are the most sensitive indexes, and the farther the deviation, the more severe the punishment. The weight values of indexes $D_5$, $D_{11}$, and $D_{15}$ changed the least, and the weights of $D_5$, $D_{11}$, and $D_{15}$ increased by 1.73%, 1.59%, and 1.76%, respectively, when the index value was reduced from eight to one, indicating that these indexes are less sensitive, and the further they deviate, the less punishment they receive.

### 4.2. Reliability Analysis

Correlation analysis compares the text evaluation results with the constant weight matter-element extension (Table 4), where *Y* is the effectiveness evaluation based on the constant weight matter-element extension method and *X* is the effectiveness evaluation based on the variable weight matter-element extension method.

The regression analysis of the evaluation results in this paper and the evaluation results of the constant weight matter-element extension method shows a strong correlation (Pearson correlation coefficient of 0.853**) and a high agreement (Spearman correlation coefficient of 0.866**) between the two. Compared with the traditional constant weight theory, using variable weight theory to calculate index weights can punish the riskier indexes, thus making the final comprehensive evaluation results closer to the engineering reality.

**Table 4.** Regression analysis of evaluation results.

| Section | Constant Weight | | Variable Weight | |
|---|---|---|---|---|
| | Grades | Results | Grades | Results |
| S1 | I | Excellent | I | Excellent |
| S2 | II | Good | IV | Poor |
| S3 | I | Excellent | I | Excellent |
| S4 | I | Excellent | I | Excellent |
| S5 | I | Good | II | Good |
| S6 | I | Excellent | I | Excellent |
| S7 | II | Good | II | Good |
| S8 | I | Excellent | I | Excellent |
| S9 | III | Medium | IV | Poor |
| S10 | II | Good | II | Good |
| Single regression: $Y = 0.504X + 0.543$, $R^2 = 0.728$ | | | | |

Based on the comparison between the evaluation results of the variable weight matter-element extension method and the currently used constant weight matter-element extension method, the overall results of the two ways are significantly consistent and correlated. However, the analysis of the evaluation results of individual sections shows differences in the evaluation results of S2 and S9, with the variable weight matter-element extension method having a poor evaluation result and the constant weight matter-element extension method having a good and medium evaluation result, respectively. The specific reasons are as follows: the single index evaluation of indexes $D_{28}$ and $D_8$ in S2 and S9 is poor, which deviates too far from the excellent interval, and these two indexes are ranked one and two, respectively, in the standing weight, and these two indexes are severely punished according to the variable weight theory, so that their weights are rapidly increased, which can also remind managers to focus on the two indexes of the road section. In concrete practice, too much reproducibility of facilities within the sample and too low a grade of function schedule can also cause a reduction in overall system effectiveness, so the effectiveness grades evaluated by the study are consistent with the judgment of concrete practice. For S1, S3, S5, S6, S7, S8, S9, and S10, the evaluation results are consistent, mainly because the weights of the indexes deviating from the excellent range of effectiveness in these eight sections are small, which have little impact on the overall evaluation results. Although certain indexes deviate from the excellent efficiency range, their punishment is minor. Therefore, the variable weight matter-element extension method can maintain the stability of the assessment results even when the non-significant indexes are in the non-optimal interval, which further indicates that the constructed model has good adaptability.

## 5. Conclusions

In this study, a comprehensive assessment of the level of cooperative efficiency of the intelligent transportation facilities on Jingxie Road, a significant road in Jinan, was conducted, and further research on the cooperative efficiency and development of the facilities was conducted using relevant indexes.

Based mainly on the system synergistic theory, this study constructs a system cooperative element model and establishes a unique index system to comprehensively assess the cooperative efficiency of the intelligent transportation facilities on JingShi Road. The evaluation model of the cooperative efficiency of intelligent transportation facilities is constructed by using the combination of variable power theory and the matter-element extension method, which reflects the dynamic influence of the change of risk index value on the efficiency, which is consistent with the specific practical work and improves the importance of the indexes with serious deviation from the efficiency. The model is better for achieving the adaptability of the evaluation object. Our findings indicate that the cooperative efficiency of different sample sections varies greatly, and the same index has other effects on different sections. Indexes such as information integration and completeness,

collaborative construction, and functional scheduling have the most significant impact on facility efficacy, and these findings can guide specific practical activities and improve the traffic environment and efficiency. In addition, issues such as further enhancing the precision of research on qualitative indexes and grades in the evaluation system and constructing models to quantify qualitative and quantitative indexes are future work to be done.

**Supplementary Materials:** The following supporting information can be downloaded at: https://www.mdpi.com/article/10.3390/su15032411/s1, Table S1: Positive index ranking; Table S2: Negative index ranking.

**Author Contributions:** Conceptualization, H.B. and L.Z.; methodology, K.L.; software, K.L.; validation, K.L.; formal analysis, X.Y.; investigation, K.L. and X.W.; resources, K.L. and H.B.; data curation, K.L.; writing—original draft preparation, K.L. and X.Y.; writing—review and editing, K.L. and X.Y.; supervision, H.B.; project administration, H.B. All authors have read and agreed to the published version of the manuscript.

**Funding:** This research was funded by Shandong Provincial Science and Technology Department, New Architecture and Key Technologies for Hybrid Augmented Intelligent "Traffic Brain" (Grant No. 2021TSGC1011).

**Institutional Review Board Statement:** Not applicable.

**Informed Consent Statement:** Not applicable.

**Data Availability Statement:** Not applicable.

**Acknowledgments:** Thanks to the reviewers and their suggestions for this work.

**Conflicts of Interest:** The authors declare no conflict of interest.

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
