# Peer review of "Cooperative Efficiency Evaluation System for Intelligent Transportation Facilities Based on the Variable Weight Matter Element Extension"

_sustainability, doi:10.3390/su15032411_

Round 1

Reviewer 1 Report

Overall, this is a good research paper with clear objective and methodology. Please spend some effort proofreading the manuscript and correcting typos. One strong suggestion is to update the references as some latest related studies are not properly cited.

Reviewer 2 Report

In this article, the authors propose a structural model of the cooperative development elements of facilities in four degrees: functional cooperation, information cooperation, business cooperation, and operation cooperation, and construct an evaluation index system of cooperative effectiveness based on the structural model. However, I will comment on some aspects to improve the quality of the manuscript, and the changes made must be highlighted:

-In Figure 1, the authors must place the source of the map, as well as its scale and coordinates, on the axes.

-Why has the setting been chosen to perform the authors' proposal?

-The authors have not included a brief introduction to the Sections that will come at the end of the Introduction Section or Related Works.

-The article must have a Related Works Section.

-Why have the authors not been cited in line 127?

-What equation are the authors talking about in line 184?

-The authors do not present the meaning of some acronyms.

-When presenting a Figure, Equation, Algorithm, Table, or Section, the authors must do so with their full words followed by the reference number. Without placing it in parentheses.

-In Lines 222, 247, 284, and 335, there is an error in the reference.

- Authors must avoid apostrophes in a scientific article.

-In line 358, the authors have referenced Figure 1, which talks about the Structural model of facility cooperative developments elements and not about the scenarios.

- The authors need to present the seven scenarios formally.

- Has the proposal presented by the authors been compared with any other theoretical or real proposal?

- In the manuscript, are there ten samples or ten sections? There needs to be more clarity of terms on the part of the authors. They need to be corrected, as it needs to be clarified for the reader.

-Authors must improve the Discussion Section.

-Authors must improve the conclusions and add future work

Reviewer 3 Report

In this study, the authors established a cooperative efficiency evaluation system and used a dynamic efficiency evaluation model based on variable weight and topological method of object elements to evaluate the cooperative efficiency of intelligent transportation facilities. Overall, the paper needs further polishing.

1. The references quoted in this paper need to be further updated. Some of the refereed journals are not well known and are not english journals.

2. The literature review needs to be rewritten. The author should comprehensively sort out and summarize the relevant research results in the past five years to highlight the contribution of this research. In addition, some old and non-classical literatures are not recommended to be kept in the paper.

3. S1, S2, S3, S4, S5, S6, S7, S8, S9, S10, what is the basis for the division of these ten samples?

4. In line 111, the serial number of the title is incorrectly marked.

5. In line 124, what is the sample size and results of key informant interviews and questionnaires? Please describe further in the paper.

6. The research method is not novel and has been used in similar areas for a long time

7. What is the number of experts invited in Section 3.1? And the selection of experts is too narrow.

8. In Section 3.3, the Pearson and Spearman models are two different methods for correlation analysis. Why does Spearman's correlation coefficient show a high agreement?

9. The rationality of the evaluation index system in Table 1 needs to be verified.

Reviewer 4 Report

The paper is really interesting, the structure is fine. I have few hints or remarks: There are two figures with number 1, please correct is! The second figure 1 shows the structural model, but I cannot find in the loop the base process, the transportation itself! I miss it.

The explanation of "Variable Weight Matter Element Extension" is too weak. Please give a more detailed description of the used model, due to the fact it is not clear for all of the readers.

I find the list of references not enough international. With only a simple search I found some relevant papers (especially on the used method).

Round 2

Reviewer 3 Report

I have no more comments.